# COVID-19′s Psychological Impact on Chronic Disease Patients Seeking Medical Care

**DOI:** 10.3390/healthcare11060888

**Published:** 2023-03-19

**Authors:** Hager Salah, AL Shaimaa Ibrahim Rabie, Amira S. A. Said, Mohammad M. AlAhmad, Ahmed Hassan Shaaban, Doaa Mahmoud Khalil, Raghda R. S. Hussein, Azza Khodary

**Affiliations:** 1Pharmaceutical Services Department, King Hamad University Hospital, Al Sayh 24343, Bahrain; 2Clinical Pharmacy Department, Faiyum Oncology Center, Fayium 63511, Egypt; alshaimaa.ph@o6u.edu.eg; 3Clinical Nutrition Department, Fayium Health Insurance Authority, Fayium 63511, Egypt; 4Clinical Pharmacy Department, College of Pharmacy, Al Ain University, Al Ain 64141, United Arab Emirates; 5Clinical Pharmacy Department, Faculty of Pharmacy, Beni-Suef University, Beni-Suef 62514, Egypt; 6Clinical Oncology Department, Faculty of Medicine, Beni-Suef University, Beni-Suef 62514, Egypt; 7Public Health and Community Medicine Department, Faculty of Medicine, Beni-Suef University, Beni-Suef 62514, Egypt; 8Clinical Pharmacy Department, Faculty of Pharmacy, October 6 University, Giza 12858, Egypt; 9Mental Health Department, Faculty of Education, Helwan University, Helwan 11795, Egypt

**Keywords:** psychological impact, DASS-21, chronic diseases, medical care

## Abstract

Background: The outbreak has harmed patients with multiple comorbidities and chronic conditions. The pandemic’s psychological impact is thought to change their routine of seeking medical care. Research Question or Hypothesis: During COVID-19, patients with chronic conditions may experience anxiety, depression, and stress, and their pattern of seeking medical care may change. Materials and Methods: In May 2021, a cross-sectional, web-based study of patients with chronic diseases was conducted. Eligible patients (1036) were assessed for psychological disorders, primarily depression, stress, and anxiety, using the DASS-21 scale, and their pattern of receiving medical care during COVID-19. Results: During the pandemic, 52.5% of the patients with chronic diseases were depressed, 57.9% were anxious, and 35.6% were stressed. Patients with chronic diseases who had moderate to severe depression (34.9% versus 45.1%, *p* = 0.001), moderate to severe anxiety (43.6% versus 53.8%, *p* = 0.001), or moderate to severe stress (14.9% versus 34.8%, *p* = 0.001) were significantly more likely to have no follow-up for their chronic conditions. Conclusions: Patients with chronic conditions experienced significant anxiety, depression, and stress during COVID-19, which changed their pattern of seeking medical care, and the majority of them did not receive follow-up for their chronic conditions.

## 1. Introduction

A cluster of pneumonia with an unknown origin was discovered in Wuhan City, Hubei Province, China, in December 2019. A new coronavirus (2019-nCoV) has been discovered as the cause of this illness [1]. The World Health Organization labeled the disease as Coronavirus Disease 2019 (COVID-19) (WHO). The new coronavirus pneumonia (COVID-19) had spread fast throughout China and the world as of 18 February 2020, resulting in thousands of confirmed cases and deaths [2]. Chronic illnesses have a high death rate and are quite costly on healthcare infrastructure [3]. The World Health Organization (WHO) estimates that in 2020, chronic diseases will account for 60% of all disease burden worldwide and 73% of all fatalities. In addition, developing nations will account for 79% of these deaths [4]. Previous studies demonstrate the high rates of stress, anxiety, and depression in chronic disease patients. It is advised that health professionals focus more on preventing and controlling these illnesses [5]. Studies reported more signs of anxiety and more stress in people with chronic disease than in those without any chronic disease during the COVID-19 pandemic [6]. Protecting older people’s mental health is crucial, especially for those who have chronic illnesses. In particular, in these difficult times that we are presently experiencing, it is necessary to provide these vulnerable segments of the population with psychological interventions and instruments aimed at enhancing their emotional and social states [7].

COVID-19 can infect persons of any age; however, older people are more susceptible to infection and have a higher fatality rate [8]. Various public health measures, such as quarantine and social isolation, have emerged in response to the COVID-19 pandemic [9]. Consequently, these measures had a negative impact on mental health, leading to a high prevalence of mental symptoms such as discomfort, anxiety, anger, loneliness, poor mood, sleeplessness, depression, and post-traumatic stress disorder [10]. These mental health side effects were attributed to stressors connected with quarantine, such as the length of the quarantine, the fear of illness or infecting others, a lack of information, and the stigma of discrimination [11]. Mental health symptoms vary from person to person depending on their thinking and sociability [12].

Patients with various comorbidities and chronic conditions such as hypertension, diabetes, renal disease, asthma, or COPD were severely affected by the COVID-19 pandemic [13], with the worst outcomes and mental health consequences (4). Patients may be avoiding medical attention out of fear of contracting the disease or as a result of quarantine [14]. This delay in obtaining treatment or omitting usual ongoing care can result in increased morbidity and mortality, which have not been considered in the assessment of the pandemic’s harm [15]. Many studies found patients with chronic conditions may be afraid to use their regular health-care services in order to reduce their chance of infection and the consequences that may result from a virus. The pandemic has significantly threatened the general public’s mental and physical health [16]. The limited access to healthcare created a huge mental burden, which results in psychological distress and anxiety disorders [17,18,19,20,21]. Patients with chronic conditions are at higher stress levels because of the higher risk of poorer COVID-19 outcomes [22]. According to the rapid spread of COVID-19 worldwide, combined with compulsory quarantine and widespread lockdowns, it triggered public fear and disseminated rumors and conspiracy theories [23].

During COVID-19, patients with chronic conditions may experience anxiety, depression, and stress, and their pattern of seeking medical care may change.

This study intends to assess the effect of COVID-19 on medical care among Egyptian patients with chronic diseases through anxiety, depression, or stress caused by the outbreak.

## 2. Materials and Methods

### 2.1. Study Design

A cross-sectional study was conducted in Egypt between March and June 2021. Approved by the Research Ethics Committee, number FMBSUREC/09052021. This study included patients with chronic diseases (diabetes, hypertension, and other chronic diseases) who received medical care in various ambulatory clinics. A total of 2176 participants were invited through text messages to participate as per government recommendations to minimize face-to-face or physical interaction as citizens continue to isolate themselves at home. Potential respondents were invited through a text message, resulting in 1450 total responses; we excluded 379 responses for not having completed data, and 35 participants did not meet the inclusion criteria. The following criteria were used to determine inclusion criteria: (1) informed consent prior to the survey; (2) residence in Egypt; (3) age 18 years or older; and (4) confirmed chronic condition diagnosis. Each participant provided information about their basic demographics as well as chronic diseases such as hypertension, diabetes mellitus, and other comorbidities. Our study aimed to investigate the following hypotheses that were more closely related to psychological impact: A higher level of anxiety, depression, and stress will be significantly associated with less regular medical follow-up for chronic disease patients during the COVID-19 pandemic in Egypt.

### 2.2. Sample Technique

An online Google form containing a questionnaire was sent via social media such as WhatsApp, Facebook, emails, and others. Respondent’s target is Egyptian adults above 18 years old with any chronic diseases. We collected data anonymously, without collecting information that could identify the respondents. The first part of the study questionnaire collected socio-demographic information, including age, gender, occupational status, city of residence, marital status, educational level, and comorbidities (diabetes, hypertension, cancer, obesity, cardiac disease, COPD, etc.).

### 2.3. Data Collection Tool

The questionnaire was translated from English to Arabic by two professionals and a native Arabic speaker with English as their first language. To evaluate the validity and reliability of the questionnaire, we performed a pilot study on 30 Egyptian participants, who were then excluded from the main study and the subsequent data analysis.

A pilot analysis was used to assess the clarity of the DASS and its appropriateness through online interviews with 30 participants. No difficulties were reported in completing it, so no further changes were made. The internal consistency of the questionnaire was assessed using Cronbach’s alpha coefficient. No interclass correlation was detected in the initial pilot study, so no components were deleted from the original version. Cronbach’s alpha for the depression domain was 0.872, that of the anxiety domain was 0.910, and that of the stress domain was 0.891.

**Part 1:** 20-item self-structured questions evaluated the socio-demographic data of study participants, including: age, gender, BMI, academic achievement, employment status, place of residence, and maternal status. In addition, data related to medical status, timing of receiving medications before and during COVID, places of getting medications, and usage of transportation vehicles were collected. The data also included whether safety measures were used while receiving medications during the pandemic or not. The questionnaire contained the status of persons for whom COVID-19 was suspected at any given time and what their response was regarding medical advice or not. Data concerning the seeking of medical advice for their chronic diseases was gathered.

**Part 2:** 21-item self-administered questions; using the DASS-21 to evaluate emotional states of anxiety, stress, and depression [24]. It is measured by the 5-point Likert scale. Final response scores were identified as normal, mild, moderate, severe, and very severe.

The depression scale assesses dysphoria, hopelessness, devaluation of life, self-deprecation, lack of interest or involvement, anhedonia, and inertia. The anxiety scale assesses autonomic arousal, skeletal muscle effects, situational anxiety, and the subjective experience of anxious affect. The stress scale is sensitive to levels of chronic non-specific arousal. It assesses difficulty relaxing, nervous arousal, being easily upset or agitated, being irritable or overly reactive, and being impatient. Scores for depression, anxiety, and stress are calculated by summing the scores for the relevant items.

The rating score was considered four choices: pick up zero when the participant saw that the choice is not applied to him at all, one when the choice is applied to him to some degree or some of the time, two when the choice is applied to him to a considerable degree or a good part of the time, and three when the choice is applied to him to very much or most of the time.

The depression score was considered normal when falling between 0 and 9, mild when falling between 10 and 13, moderate when falling between 14 and 20, severe when falling between 21 and 27, and extremely severe when falling at 28 or above. The anxiety score was considered normal when it was between 0 and 7, mild when it was between 8 and 9, moderate when it was between 10 and 14, severe when it was between 15 and 19, and extremely high when it was 20 or above. The stress score was considered normal when it fell between 0 and 14, mild when it fell between 15 and 18, moderate when it fell between 19 and 25, severe when it fell between 26 and 33, and extreme when it fell between 34 and above.

Using Epi Info StatCalc [25], the sample size for a population survey was calculated at a 95% confidence level with a 5% acceptable margin of error, one design effect, and 50% expected frequency (of regular follow-up or a positive DASS). The minimum sample size was found to be at least 384 people, which was tripled to overcome the selection bias.

### 2.4. Statistical Analysis

The Statistical Package for Social Science (SPSS) version 25 was used to gather, code, and analyze the data (IBM, USA) IBM Corp. Released 2017. IBM SPSS Statistics for Windows, Version 25.0. Armonk, NY: IBM Corp. We estimated the frequency distribution of categorical variables as a percentage and the mean and SD for scale variables. We categorized the scale variables by median (age at less than or equal to 32 and more than 32 years, and BMI at less than or equal to 27.8 and more than 27.8). The Chi-Square Test of Independence was utilized to determine a connection between categorical variables (difference between follow-up and no follow-up and age, sex, residence, working status, occupation, education, chronic disease, degree categories of depression, anxiety, and stress). Binary logistic regression was used to identify the determinants of no follow-up among the hypothesized factors that can affect the probability of its occurrence. The mentioned binary logistic model is the best model that explained the probability of no follow-up occurrence after excluding intercorrelation between variables and redundant variables such as BMI, working status, marital status, and residence. *p* values of ≤0.05 were considered significant.

## 3. Results

The total number of eligible responses was 1036 patients with chronic diseases. They were filling on their behalf and were included. The baseline characteristics of chronic disease patients are shown in Table 1, with a median age of 32, a marriage rate of 52.4%, and a majority having more than one chronic disorder (i.e., hypertension plus diabetes) at around 37.5%. Diabetes and hypertension were the most common chronic diseases in our sample population, but we also included other comorbidities (cancer, obesity, COPD, cardiac disease, and autoimmune disease); however, they were not significant in our sample population.

In addition, Table 2 showed the information about COVID-19 infection status and medical treatment received for it. 59.3% were clinically suspected of having COVID-19, 35.8% were self-isolated, and 32.9% went to the hospital.

Moreover, the patients’ follow-up pattern before and during the COVID-19 pandemic was illustrated in Table 3. Our results revealed that 73.6% were regularly collecting their medication before the COVID-19 pandemic and dropped to 43.5% during the COVID-19 pandemic as 63.2% had a fear of COVID-19 infection.

Furthermore, the results illustrated that 52%, 60%, and 35.6% of patients with chronic diseases suffered from depression, anxiety, and stress, ranging from mild to very severe, respectively, as shown in Table 4.

The univariate analysis revealed the following statuses: being female, being younger, having a low BMI, being unmarried, having a low educational level, not working, having an urban residency, and not preferring telemedicine were significantly associated with less regular follow-up, as illustrated in Table 5. While having DM plus hypertension was more significantly associated with follow-up.

DASS-21 was used to evaluate the emotional states of anxiety, stress, and depression, all of which were significantly associated with regular follow-up.

The results illustrated that after adjustment for age, gender, residence, presence of depression, presence of anxiety, and presence of stress caused by the COVID-19 pandemic, it was found that the presence of anxiety caused by the COVID-19 pandemic increased the probability of no follow-up (in other words, the stress caused by the COVID-19 pandemic decreases the follow-up rate) with OR, the 95% CI of OR was 2.693, 1.856 to 3.908 as indicated in Table 6. In addition, being old and male decreased the probability of no follow-up significantly with OR; the 95% CI of OR was 0.318, 0.236 to 0.428, and 0.608, 0.450 to 0.822 for age and sex, respectively.

## 4. Discussion

The global healthcare system is being stressed by the coronavirus disease 2019 (COVID-19) pandemic [26]. This study aimed to evaluate the psychological impact of the COVID-19 pandemic on patients with chronic conditions who may have suffered from anxiety, depression, and stress during COVID-19, which may have affected their pattern of seeking medical care among the Egyptian population [27]. Healthcare administrators, emergency responders, and healthcare clinicians must all receive coaching and education on psychological issues from the healthcare system [28]. Identifying, establishing, and allocating evidence-based resources for disaster-related mental health, psychological well-being crises and referral, particular patient needs, and alarm and distress treatment are all tasks that mental health and emergency response systems must collaborate on [29]. Despite health issues, medical treatment professionals eventually have a vital role in identifying psychosocial requirements and providing psychosocial aid to their patients, as well as social efforts that should be incorporated into overall pandemic healthcare. A rise in known risk factors for mental health issues has been attributed to COVID-19. Quarantine and physical isolation are also present, along with oddities and discomfort [30]. This study revealed that 52.2% of patients did not follow-up regularly with their chronic diseases during COVID-19; 63.2% of the patients attributed the absence of follow-up to their fear of COVID-19 infection, 21.3% of the patients attributed the no follow-up status to the cost of medical care with limited resources during COVID-19; and 58% preferred to follow-up with telemedicine. New techniques for providing care through telemedicine to lessen in-person interactions.

To enable health care clinicians to keep scheduled appointments, new digital and virtual healthcare practices must be used, in accordance with a previous study [31]. Additionally, the usage of apps can aid in the self-management of chronic illnesses, such as diabetes, where continuous glucose monitoring is possible. However, the bulk of those suffering from non-communicable diseases reside in low- and middle-income nations [32]. Our findings showed that the fear enveloping people’s thoughts about the pandemic and the hazards of becoming infected by stepping outside was the main reason for the absence of medical follow-up in chronic disease patients. About half of individuals with medical illnesses handled their conditions by calling doctors through telemedicine and collecting their own medication from a community pharmacy. Previous studies revealed that around 55% of patients with chronic diseases did not contact their doctors and depended on self-medication [33,34]. In concordance with our findings, previous studies showed that people have generally been practicing—or have been pushed to practice—rational medical practices in the face of the greater concern consuming their minds regarding the pandemic and the risks of contracting it by venturing outside. The majority of participants with medical illnesses controlled their tolerable suffering by following the medications already provided or by calling doctors as necessary. Only a true emergency (fracture) or a perceived emergency (illness) had prompted the travel to a medical facility away from home (suspected COVID-19) [33].

Through timely detection, referral, and care of suspected cases, community pharmacies and pharmacy employees play a critical role in avoiding the “community transmission” stage of COVID-19. Yet, in accordance with government guidelines, our study revealed most patients were aware of self-care to avoid infection transmission, including hand rubs with alcohol for 68.9% of patients and proper use of face masks for 92.5% of patients [35].

Moreover, this study found that a low educational level was significantly associated with no follow-up, as was urban residence, which was more significantly associated with no follow-up. In these times, the socioeconomic division, combined with limited access to high-quality health care, has become even more apparent [36]. On the other hand, many people have limited access to the internet, so teleconsultation would be difficult for them. This may have played a factor due to the reduced study sample size and some target people not receiving the survey, which results in a limitation in our study. Apart from the socioeconomic divide highlighting poor access to health care and advice, the pandemic resulted in the emergence of stress, fear, and anxiety disorders across the population, regardless of social status [37]. As a result, COVID-19 has increased the prevalence of mental health issues, as has been the case in the past following novel disease epidemics and natural disasters. Not just COVID-19, but all significant emergencies surely result in mental health issues. Studies of previous outbreaks revealed that 31.2% of people quarantined due to COVID-19 in Toronto, Canada, and roughly 35% of SARS survivors in Hong Kong both experienced symptoms of anxiety and/or depression [23,38].

Using the DASS-21 tool, we discovered that 45.1% of patients with chronic diseases had moderate to severe depression, 53.8% had moderate to severe anxiety, and 34.8% had moderate to severe stress. A univariate analysis revealed that the more severe the depression, anxiety, and stress, the more severe the disease. We found that the greater the increase in the scores of depression, anxiety, and stress, the more they were significantly associated with the no follow-ups, which matches with previous studies [39,40].

There is a need to raise awareness among chronic disease patients, particularly among the poor, about the significance of sticking to their medications [41]. Patients with chronic conditions, particularly those from poor backgrounds, need to be made aware of the value of taking their prescriptions as prescribed. It would be wise to keep in mind the tremendous patient population of so many other diseases, especially the chronic diseases, which need regular monitoring, advice, and medications. Although the public resources at the moment are primarily focused on overcoming the huge challenge of containing the COVID pandemic and looking for effective therapies. To reduce overall concern and provide the needed incentive for community health promotion, additional proactive steps such as creating consultation facilities or streamlining the prescription refill procedure for such individuals will be helpful [41].

The relevant contribution of this study to the field of literature is the urgency of regular monitoring and providing patients with “counseling for patients,” especially those suffering from chronic diseases, to help them overcome any fear during any pandemic and control their diseases well.

## 5. Limitations

The study should be conducted on larger scales in different countries as a multicentered study. Also, the study should be well designed to avoid any bias during the sampling procedure. More comorbidities must be evaluated and compared.

## 6. Conclusions

Though public resources are focused on overcoming the herculean task of containing the COVID-19 pandemic and finding effective therapies, it is prudent to remember that the vast patient population of many other diseases, particularly chronic diseases, requires regular monitoring, advice, and medication. More proactive steps, such as providing consultation services or making the procedure of refilling medicines for such patients easier, can help alleviate anxiety in general and provide the necessary impetus for community health promotion.

## Figures and Tables

**Table 1 healthcare-11-00888-t001:** Baseline characteristics of participants.

Characteristics	Number (%)
**Age**	
Mean ± SD	32.8 ± 12.8
Median	32.00
Young (≤32)	557 (53.8)
Old (>32)	479 (46.2)
**Sex**	
Female	448 (43.2)
Male	588 (56.8)
**BMI**	
Mean ± SD	27.3 ± 4.5
Median	27.8
Low (≤28))	559 (54.0)
High (>28)	477 (46.0)
**Marital status**	
Widowed	12 (1.1)
Single/NA	473 (45.7)
Married	543 (52.4)
Divorced	8 (0.8)
**Educational level**	
Bachelor	337 (32.5)
Intermediate Technical education	12 (1.1)
Post-graduation	365 (35.3)
Student (high school or faculty)	305 (29.5)
Not educated	17 (1.6)
**Occupation**	
Non-medical personnel	426 (41.1)
Student	342 (33.0)
On pension	24 (2.3)
Not working	49 (4.7)
Medical personnel	195 (18.9)
**Residence**	
Urban	143 (13.8)
Rural	893 (86.2)
**Chronic disease**	
DM	120 (11.6)
HTN	85 (8.2)
DM and HTN	184 (17.8)
Others/Multiple co-morbidities	647 (62.4)

SD = standard deviation; DM = diabetes mellitus; HTN = hypertension.

**Table 2 healthcare-11-00888-t002:** COVID-19 infection status and medical treatment received.

Item	Number (%)
**Did you have clinically suspected COVID-19**	
No	422 (40.7)
Yes	614 (59.3)
**Seeking medical care**	
Nearest pharmacy	115 (18.7)
Go to hospital	202 (32.9)
At home by doctor	97 (15.8)
Telemedicine	178 (29)
self-isolated	22 (3.58)

**Table 3 healthcare-11-00888-t003:** Patients’ follow-up patterns before and during the COVID-19 pandemic.

Item	Number (%)
**Medication collection regularly before COVID-19**	
Not regularly	274 (26.4)
Yes, regularly	762 (73.6)
**Medication collection regularly during COVID-19**	
Not regularly	585 (56.4)
Yes, regularly	451 (43.5)
**Place to collect medication**	
Health Insurance Org	276 (26.6)
Community pharmacy	570 (55.0)
No medication collection	128 (12.4)
University hospital	28 (2.7)
General hospital	34 (3.3)
**Transportation to site of medication collection**	
No	634 (61.2)
Yes	402 (38.8)
**Seeking medical care for chronic disease during COVID-19**	
Monthly	290 (28.0)
Every 3 months	167 (16.1)
Every 6 months	38 (3.7)
Not follow-up my chronic disease	541 (52.2)
**Causes of not follow-up medical care (n = 541)**	
Cost of medical care with limited resources during COVID	115 (21.3)
Fear of COVID infection	342 (63.2)
Far site of medical care	53 (9.8)
Cannot find who follow me	31 (5.7)
Do you prefer telemedicine	
No	435 (42.0)
Yes	601 (58.0)
**Use mask during medication collection**	
No	78 (7.5)
Yes	958 (92.5)
**Rub your hand with Alcohol during medication collection**	
No	322 (31.1)
Yes	714 (68.9)
**Only social distancing during medication collection** **(Without face mask or alcohol rub)**	
No	837 (80.8)
Yes	199 (19.2)

**Table 4 healthcare-11-00888-t004:** DASS-21 score among participants.

Items	Number (%)
**Depression**	
Normal	492 (47.5)
Mild	127 (12.3)
Moderate	164 (15.8)
Severe	132 (12.7)
Very severe	121 (11.7)
**Anxiety**	
Normal	436 (42.1)
Mild	93 (9.0)
Moderate	241 (23.3)
Severe	100 (9.7)
Very severe	166 (16.0)
**Stress**	
Normal	667 (64.4)
Mild	107 (10.3)
Moderate	100 (9.7)
Severe	112 (10.8)
Very severe	50 (4.8)

**Table 5 healthcare-11-00888-t005:** Univariate analysis for risk factors associated with less regular follow-up of chronic disease during the COVID-19 pandemic.

Risk Factors	Follow-Up (no = 495)	No Follow-Up (no = 541)	*p*-Value	Comments
**Sex**			<0.001 *	Female sex is associated with less regular follow-up
Female	147 (29.7%)	301 (55.6%)		
Male	348 (70.3%)	240 (44.4%)		
**Age**			<0.001 *	Youngers age is associated with less regular follow-up
Young	197 (39.8%)	360 (66.5%)		
old	298 (60.2%)	181 (33.5%)		
**BMI**			<0.001 *	Low BMI is associated with less regular follow-up
Low (≤28)	204 (41.2%)	355 (65.6%)		
High (>28)	291 (58.8%)	186 (34.4%)		
Marital status			<0.001 *	Unmarried isassociated with less regular follow-up
Unmarried	151 (30.5%)	342 (63.2%)		
Married	344 (69.5%)	199 (36.8%)		
**Educational level**			<0.001 *	Low educational level is associated with less regular follow-up
Bachelor	182 (36.8%)	155 (28.7%)		
Intermediate	7 (1.4%)	5 (0.9%)		
Post-graduation	213 (43.0%)	152 (28.1%)		
Student	85 (17.2%)	220 (40.7%)		
Not educated	8 (1.6%)	9 (1.7%)		
**Educational level**			<0.001*	
Till secondary	100 (20.2%)	234 (43.3%)		
University and post	395 (79.8%)	307 (56.7%)		
**Occupation**			<0.001 *	Not-working participants were more likely to be less regular with follow-up
Non-medical personnel	252 (50.9%)	174 (32.2%)		
Student	92 (18.6%)	250 (46.2%)		
On pension	21 (4.2%)	3 (0.6%)		
Not working	23 (4.6%)	26 (4.8%)		
Medical personnel	107 (21.6%)	88 (16.3%)		
**Working status**			<0.001 *	
Not working	136 (27.5%)	279 (51.6%)		
Working	359 (72.5%)	262 (48.4%)		
Residence			0.026 *	Urban residence was more associated with less regular follow-up
Urban	56 (11.3%)	87 (16.1%)		
Rural	439 (88.7%)	454 (83.9%)		
**Chronic disease**			<0.001 *	DM with HTN was more associated with follow-up
DM	51 (10.3%)	69 (12.8%)		
HTN	47 (9.5%)	38 (7.0%)		
DM and HTN	128 (25.9%)	56 (10.4%)		
Others	269 (54.3%)	378 (69.9%)		
**Prefer telemedicine**			0.046 *	Those who do not prefer telemedicine were more likely not to regularly follow-up
No	192 (38.8%)	243 (44.9%)		
Yes	303 (61.2%)	298 (55.1%)		
**Depression**			<0.001 *	Increase the score of depression; increase the less regular follow-up
Normal	291 (58.8%)	201 (37.2%)		
Mild	31 (6.3%)	96 (17.7%)		
Moderate	68 (13.7%)	96 (17.7%)		
Severe	80 (16.2%)	52 (9.6%)		
Very severe	25 (5.1%)	96 (17.7%)		
**Depression**			0.001 *	
Normal to mild	322 (65.1%)	297 (54.9%)		
Moderate to very severe	173 (34.9%)	244 (45.1%)		
**Anxiety**			<0.001 *	Increase the score of anxiety; increase the less regular follow-up
Normal	266 (53.7%)	170 (31.4%)		
Mild	13 (2.6%)	80 (14.8%)		
Moderate	109 (22.0%)	132 (24.4%)		
Severe	53 (10.7%)	47 (8.7%)		
Very severe	54 (10.9%)	112 (20.7%)		
**Anxiety**			0.001 *	
Normal to mild	279 (56.4%)	250 (46.2%)		
Moderate to very severe	216 (43.6%)	291 (53.8%)		
**Stress**			<0.001 *	Increase the score of stress; increase the less regular follow-up
Normal	366 (73.9%)	301 (55.6%)		
Mild	55 (11.1%)	52 (9.6%)		
Moderate	32 (6.5%)	68 (12.6%)		
Severe	33 (6.7%)	79 (14.6%)		
Very severe	9 (1.8%)	41 (7.6%)		
**Stress**			0.001 *	
Normal to mild	421 (85.1%)	353 (65.2%)		
Moderate to very severe	74 (14.9%)	188 (34.8%)		

* *p*-value is significant. Chi-Squared test

**Table 6 healthcare-11-00888-t006:** Multivariable binary logistic regression analysis for prediction of risk factors associated with no follow-up of chronic diseases during the COVID-19 pandemic.

Independent Variables	*p*-Value	OR	95% C.I. for OR	
			Lower	Upper
Old age (>32)	**0.001 ***	0.318	0.236	0.428
Male sex	0.001 *	0.608	0.450	0.822
Rural residence	0.834	1.044	0.700	1.556
Presence of depression	0.704	1.080	0.727	1.605
Presence of anxiety	0.001 *	2.693	1.856	3.908
Presence of stress	0.703	1.076	0.738	1.570

OR = Odds ratio CI = confidence interval * *p*-value is significant.

## Data Availability

The data will be available from the corresponding author upon request.

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
