# Peer review of "COVID-19′s Psychological Impact on Chronic Disease Patients Seeking Medical Care"

_healthcare, 2023, doi:10.3390/healthcare11060888_

Round 1
Reviewer 1 Report
Thank you for the possibility to review this article.
There are several typos and grammar error that should be adressed.
"pandemic .However", "variy from person to person depending on their thinking and so-52 ciability(5).", "consequences(4) . Patients", "virus.The", "Table (3) .Our", "of 32, 52.4", "percent,60 percent", and so on.
In addition, the work should be more compliant. For example, some times the authors use the word "percentage" and other times the symbol "%."; "COVID", "COVID-19", and so on.
Aim is poorly described and not very clear. Moreover, one or more hypothesis should be reported.
Sample
Sample information is laking.
How many subjects were administered the instruments? How many instruments were collected? How many incomplete and not utilized? Was there an initial list of subjects? How many subjects were on the initial list? What were the characteristics of the subjects on the initial list? What kind of sampling was done? What were the sociodemographic characteristics of the subjects? Etc. Etc.
Instruments
The authors did not use a validated version of DASS in their language but translated the tool into their language. This is not methodologically correct. In any case, if they decided to translate the instrument themselves they should do an initial validation of it at least with an exploratory factor analysis and the respective Crombach's alphas for each scale. The limitations of the study should also report these shortcomings.
The DASS should be described more accurately, reporting information on scales, calculation of scores, examples of items, etc.
Data Analysis
The authors used a Student's t but do not report indications on verification of assumptions to perform this analysis, such as normality of the curve, homoschedasticity, etc.
Results
all acronyms should be explained (DM, HTM, and so on), at least in the table notes.
The authors say "The independent student T test was used to discover the difference between two categories, and the Chi-Square Test of Independence was utilized to determine a connection between categorical variables. Binary ogistic regression was used to identify the determinants of no follow up.", but they should be clearer and say with which variables each of these tests were used. Also, the same information should be given in the text and tables. While the text should also state more clearly the statistical parameters (e.g., Student's t-value, degrees of freedom, etc.).
Rationale.
the authors say "This study aimed to evaluate Psychological impact of COVID-19 pandemic on Patients with chronic conditions who may suffered from anxiety, depression, and stress during COVID-19, and affected their pattern of seeking medical care among Egyptian population." but the introduction does not adequately introduce this objective. The authors should better justufucate why their proposed study is useful by starting from the literature and any gaps in the literature. The discussion should better focus on the results against these gaps.
I suggest the following papers:
Taylor, S. (2021). COVID stress syndrome: Clinical and nosological considerations. Current psychiatry reports, 23, 1-7.
Renati, R., Bonfiglio, N. S., & Rollo, D. (2023). Italian University Students’ Resilience during the COVID-19 Lockdown—A Structural Equation Model about the Relationship between Resilience, Emotion Regulation and Well-Being. European Journal of Investigation in Health, Psychology and Education, 13(2), 259-270.
Lakhan, R., Agrawal, A., & Sharma, M. (2020). Prevalence of depression, anxiety, and stress during COVID-19 pandemic. Journal of neurosciences in rural practice, 11(04), 519-525.
Overall
the work is unclearly written. It is not clear what the authors actually want to demonstrate and why they chose to use certain instruments and variables. The study is not adequately justified in the introductory section. The work seems incomplete and not mature. The various limitations of the study should be reported in the limitations of the paper.
Author Response
|
|
||
|
Comment |
Author Response |
Text Insertion (if applicable)/ section
|
|
1. There are several typos and grammar error that should be adressed. |
Done English editing certificate attached |
|
|
2. In addition, the work should be more compliant. For example, some times the authors use the word "percentage" and other times the symbol "%."; "COVID", "COVID-19", and so on |
Done |
|
|
3. Aim is poorly described and not very clear |
Line 94-97
Our study aimed to investigate the following hypotheses that were more closely related to psychological impact, A higher level of anxiety ,depression and stress will be significantly associated with less regular medical follow up for chronic disease patients during COVID-19 pandemic in Egypt .
|
|
|
4. one or more hypothesis should be reported |
Line 76-78 Research Question or Hypothesis, During COVID-19, patients with chronic conditions may experience anxiety, depression, and stress, and their pattern of seeking medical care may change. |
|
|
5. Sample information is laking. How many subjects were administered the instruments? How many instruments were collected? How many incomplete and not utilized? Was there an initial list of subjects? How many subjects were on the initial list? What were the characteristics of the subjects on the initial list? What kind of sampling was done? What were the sociodemographic characteristics of the subjects? Etc. Etc.
|
Line 85-90 20176 Subject was invited through text messages to participate as per recommendations government to minimize face-to-face or physical interaction as citizens continue to isolate themselves at home. Potential respondents were invited through a text message result in 1450 total response, we exclude 379 responses not completed data, and 35 not met the inclusion criteria. Line 101-105 We collected data anonymously, without collecting information that could identify the respondents. The first part of the study questionnaire collected socio-demographic information include age, gender, occupational status, city of residence, Marital status, educational level and comorbidities include diabetes ,hypertension,cancer,obesity, cardiac disease, COPD and others |
|
|
6. The authors did not use a validated version of DASS in their language but translated the tool into their language. This is not methodologically correct. In any case, if they decided to translate the instrument themselves they should do an initial validation of it at least with an exploratory factor analysis and the respective Crombach's alphas for each scale. The limitations of the study should also report these shortcomings. The DASS should be described more accurately, reporting information on scales, calculation of scores, examples of items, etc.
|
We conducted the validation but unfortunately we didn’t mentioned it in the methodology so, we put it in details again. Line 107-117 Line 130-149 |
|
|
Data Analysis 1. The authors used a Student's t but do not report indications on verification of assumptions to perform this analysis, such as normality of the curve, homoschedasticity, etc |
Line 157-168 |
|
|
Results all acronyms should be explained (DM, HTM, and so on), at least in the table notes. The authors say "The independent student T test was used to discover the difference between two categories, and the Chi-Square Test of Independence was utilized to determine a connection between categorical variables. Binary Logistic regression was used to identify the determinants of no follow up.", but they should be clearer and say with which variables each of these tests were used. Also, the same information should be given in the text and tables. While the text should also state more clearly the statistical parameters (e.g., Student's t-value, degrees of freedom, etc.).
|
Add in table 1 note Added in table 5 note Chi-Square Test |
|
|
Rationale. the authors say "This study aimed to evaluate Psychological impact of COVID-19 pandemic on Patients with chronic conditions who may suffered from anxiety, depression, and stress during COVID-19, and affected their pattern of seeking medical care among Egyptian population." but the introduction does not adequately introduce this objective. The authors should better justufucate why their proposed study is useful by starting from the literature and any gaps in the literature. The discussion should better focus on the results against these gaps. I suggest the following papers: Taylor, S. (2021). COVID-19 stress syndrome: Clinical and nosological considerations. Current psychiatry reports, 23, 1-7. Renati, R., Bonfiglio, N. S., & Rollo, D. (2023). Italian University Students’ Resilience during the COVID-19 Lockdown—A Structural Equation Model about the Relationship between Resilience, Emotion Regulation and Well-Being. European Journal of Investigation in Health, Psychology and Education, 13(2), 259-270. Lakhan, R., Agrawal, A., & Sharma, M. (2020). Prevalence of depression, anxiety, and stress during COVID-19 pandemic. Journal of neurosciences in rural practice, 11(04), 519-525.
|
We add references from literature highlighted |
|

Reviewer 2 Report
The content of his study is insignificant, and more so in 2023; the theme, the recruitment of the sample, the reliability, etc. A sample selection based on facebook, wttps, is not adequate, on the other hand... how many subjects were sent the survey? as well as why they indicate diabetes and not other disorders. Your proposal is correct, although not adequate for a publication of impact.
Author Response
|
|
||
|
Comment |
Author Response |
Text Insertion (if applicable)/ section
|
|
The content of this study is insignificant, and more so in 2023,the theme, the recruitment of the sample,the reliability,etc. Asample selection based on facebook,wttps,is not adequate, on the other hand,how many subjects were sent the survey,? As well as why they indicate diabetes and not other disorders. Your proposal is correct,although not adequate for a publication of impact. |
Line 296-299 The relevant contribution of this study to the field of literature is the urgency of regular monitoring and providing patients 'counselling for patients especially those suffering from chronic diseases to help them to overcome any fear during any pandemic to control their diseases well
Line 85-90 20176 Subject was invited through text messages to participate as per recommendations by governoment to minimize face-to-face or physical interaction as citizens continue to isolate themselves at home. Potential respondents were invited through a text message result in 1450 total response, we exclude 379 responses not completed data, and 35 not met the inclusion criteria Line 174-176 Diabetes and hypertensions the most highest percentage chronic diseases on our sample population, we include other comorbidities eg., cancer, obesity , COPD, Cardiac disease and autoimmune disease but not significant in our sample population |
|

Reviewer 3 Report
Comments and suggestions
It was my pleasure to review this manuscript dealing with COVID-19's psychological impact on chronic disease patients seeking medical care in Egypt. This manuscript aims to investigate the psychological impact of the Covid-19 pandemic on patients with chronic illnesses who may have experienced anxiety, depression, or stress during the outbreak, as well as the effect of COVID-19 on their habit of seeking medical care among Egyptians. A cross-sectional study was conducted in Egypt, and an online google forum containing a questionnaire was sent via social media such as WhatsApp, Facebook, and others. In brief, I found the topic quite interesting. But with the sole objective of improving the quality of the manuscript, I will allow myself to make a few comments:
1. This manuscript has a well-structured description, but I still suggest this manuscript should undergo extensive English revisions. Because many minor mistakes exist and should be corrected.
2. The Introduction section contains numerous sentences that refer to a single bibliographic citation. These statements must therefore be enriched with further references drawing from the current rich international bibliography.
3. Materials and Methods
This study collected data from social media platforms such as WhatsApp, Facebook, and others, using web-based questionnaires. Although this study performed the calculation of sample size in advance and settled up inclusion criteria such as informed consent prior to the survey, residence in EGYPT, and age 18 years or older,
The results may not be representative and exist as a question of external validity. Convenience samples are quite prone to research bias. Since the researcher draws the sample based on convenience and not equal probability, convenience samples never result in a statistically balanced selection of the population. This leads to sampling bias. This part should mention in the study limitation.
4. Results
Table 1: The percentage of many variables such as “Marital status”, “Educational level”, “Occupation”, and “Chronic disease” didn’t equal 100%. Please confirm it.
Table 1: Educational level should address more specifically or have an explanation. For example, in your study “Student” educational level indicated what kind of educational level? The author needs to mention this part in order to let readers can understand it.
Table 2: Seeking medical care variable
The percentage of the self-isolated category didn’t correct. 35.8% should be corrected to 3.58%.
Table 3: Causes of not follow-up medical care variable (no=)
Please add the number in the “Causes of not follow up medical care” variable. The status in present revealed an uncompleted sentence.
Table 4: Please delete the word of “P-value is significant” (page 6).
5. Discussion
Lines 175-176: Around 55% of patients with chronic diseases did not contact their doctors and depended on self-medication (16,17). This result came from your study or came from two previous studies. Please confirm it.
Line 181: Proper use of face masks for about 92.5 % of patients (18). This result came from your study or came from two previous studies. Please confirm it.
Line 185: Many people did not have access to the internet, so teleconsultation would be difficult for them. This study collected data from social media platforms and the age of participants is the median age of 32. Why many people did not have access to the internet? Please explain your opinion and add this important information to the revised manuscript.
Author Response
|
|
||
|
Comment |
Author Response |
Text Insertion (if applicable)/ section
|
|
1. This manuscript has a well-structured description, but I still suggest this manuscript should undergo extensive English revisions. Because many minor mistakes exist and should be corrected.
|
Done English editing certificate attached |
|
|
2. The Introduction section contains numerous sentences that refer to a single bibliographic citation. These statements must therefore be enriched with further references drawing from the current rich international bibliography. |
Done and add more reference |
|
|
3. Materials and Methods This study collected data from social media platforms such as WhatsApp, Facebook, and others, using web-based questionnaires. Although this study performed the calculation of sample size in advance and settled up inclusion criteria such as informed consent prior to the survey, residence in EGYPT, and age 18 years or older, The results may not be representative and exist as a question of external validity. Convenience samples are quite prone to research bias. Since the researcher draws the sample based on convenience and not equal probability, convenience samples never result in a statistically balanced selection of the population. This leads to sampling bias. This part should mention in the study limitation.
|
Line 302-304 The Study limitations: The study should be conducted on larger scales in different countries as multicentered study. Also, the study should be well designed to be away from any bias during the sampling procedure. More comorbidities must be evaluated and compared.
|
|
|
4. Results Table 1: The % of many variables such as “Marital status”, “Educational level”, “Occupation”, and “Chronic disease” didn’t equal 100%. Please confirm it.
|
Done highlighted in table |
|
|
Table 1: Educational level should address more specifically or have an explanation. For example, in your study “Student” educational level indicated what kind of educational level? The author needs to mention this part in order to let readers can understand it.
|
Student ( high school or faculty) added in table 1 |
|
|
Table 2: Seeking medical care variable The % of the self-isolated category didn’t correct. 35.8% should be corrected to 3.58%.
|
Corrected in table 2 |
|
|
Table 3: Causes of not follow-up medical care variable (no= Please add the number in the “Causes of not follow up medical care” variable. The status in present revealed an uncompleted sentence.
|
NO=514 added in table 3 |
|
|
Table 4: Please delete the word of “P-value is significant” (page 6).
|
Done |
|
|
. Discussion Lines 175-176: Around 55% of patients with chronic diseases did not contact their doctors and depended on self-medication (16,17). This result came from your study or came from two previous studies. Please confirm it.
|
Line 248 Previous studies revealed around 55%
|
|
|
Line 181: Proper use of face masks for about 92.5 % of patients (18). This result came from your study or came from two previous studies. Please confirm it.
|
Our study revealed Line 260
|
|
|
Line 185: Many people did not have access to the internet, so teleconsultation would be difficult for them. This study collected data from social media platforms and the age of participants is the median age of 32. Why many people did not have access to the internet? Please explain your opinion and add this important information to the revised manuscript.
|
Line 95-90 20176 Subject was invited through text messages to participate as per recommendations by government to minimize face-to-face or physical interaction as citizens continue to isolate themselves at home. Potential respondents were invited through a text message result in 1450 total response, we exclude 379 responses not completed data, and 35 not met the inclusion criteria Line 267-270 On the other hand Many people did not have access to the internet, so teleconsultation would be difficult for them and this reduce study sample size and some target people not received the survey which may m made misconduct and limitation in our study |
|

Reviewer 4 Report
Thank you for your paper.
The special situation of people with chronic illnesses during COVID 19 is of particular interest, both for professionals and for the patients and their care-takers.
However, your paper must be significantly improved in order to bring a relevant contribution to the field.
The sample size is very good, it allows a lot of study on the data.
The theoretical frame must be enriched, more relevant literature must be cited, the particular situation from your country for chronically ill patients should be presented.
The statistical part should be completed and discussed in extended form.
It would be interesting to see comparisons of attitudes for different chronic illnesses.
The largest part of the results are dedicated to the descriptive part of the demographic analysis. Accent should be put on the statistical part, more complex than simple percentages.
The discussion part needs to be improved and completed based on the statistical data obtained.
In the current for the paper does not bring relevant contribution to the field literature.
Author Response
|
|
||
|
Comment |
Author Response |
Text Insertion (if applicable)/ section
|
|
Thank you for your paper. The special situation of people with chronic illnesses during COVID-19 19 is of particular interest, both for professionals and for the patients and their care-takers.
|
Thanks for the nice words |
|
|
However, your paper must be significantly improved in order to bring a relevant contribution to the field.
|
Done after our attempt to respond to all of the comments |
|
|
The sample size is very good, it allows a lot of study on the data. |
Thanks |
|
|
The theoretical frame must be enriched, more relevant literature must be cited, the particular situation from your country for chronically ill patients should be presented.
|
Done highlighted |
|
|
The statistical part should be completed and discussed in extended form.
|
Done add more reference |
|
|
It would be interesting to see comparisons of attitudes for different chronic illnesses.
|
Thanks for the appreciated point of view, this can be considered for future studies |
|
|
The largest part of the results are dedicated to the descriptive part of the demographic analysis. Accent should be put on the statistical part, more complex than simple %s.
|
We described the baseline and the DASS parameters then we run the inferential statistics according to our outcome of interest. |
|
|
The discussion part needs to be improved and completed based on the statistical data obtained.
|
Done highlighted |
|
|
. In the current for the paper does not bring relevant contribution to the field literature.
|
Line 296-299 The relevant contribution to the field of literature is the urgency of regular monitoring and providing patients 'counselling for patients especially those suffering from chronic diseases to help them to overcome any fear during any pandemic to control their diseases well.
|
|
|
The minimum word count is 4000 Our word count is 2880
|
Done more than 4000 word
|
|

Round 2
Reviewer 1 Report
No other comments
Author Response
Thanks so much.
Reviewer 2 Report
The content of this study is insignificant, and more so in 2023,the theme, the recruitment of the sample,the reliability,etc. A sample selection based on facebook,wttps,is not adequate, on the other hand,how many subjects were sent the survey,?
As well as why they indicate diabetes and not other disorders.
Your proposal is correct,although not adequate for a publication of impact.
Author Response
|
|
||
|
Comment |
Author Response |
Text Insertion
|
|
Round 1 The content of this study is insignificant, and more so in 2023,the theme, the recruitment of the sample,the reliability,etc. Asample selection based on facebook,wttps,is not adequate, on the other hand,how many subjects were sent the survey,? As well as why they indicate diabetes and not other disorders. Your proposal is correct,although not adequate for a publication of impact. Round 2 The content of this study is insignificant, and more so in 2023,the theme, the recruitment of the sample,the reliability,etc. A sample selection based on facebook,wttps,is not adequate, on the other hand,how many subjects were sent the survey,?
As well as why they indicate diabetes and not other disorders. Your proposal is correct,although not adequate for a publication of impact.
|
We previously answer these comments in round 1 Line 308-311 The relevant contribution of this study to the field of literature is the urgency of regular monitoring and providing patients 'counselling for patients especially those suffering from chronic diseases to help them to overcome any fear during any pandemic to control their diseases.
Sample selection depending on social media due to circumstances of quarantine during Covid-19 and all types of social media widely used during pandemic in urban and rural area and this safest way to communicate during pandemic.
Line 97-102 20176 Subject was invited through text messages to participate as per recommendations by governoment to minimize face-to-face or physical interaction as citizens continue to isolate themselves at home. Potential respondents were invited through a text message result in 1450 total response, we exclude 379 responses not completed data, and 35 not met the inclusion criteria.
Line 186-188 Diabetes and hypertensions the most highest percentage chronic diseases on our sample population, we include other comorbidities eg., cancer, obesity , COPD, Cardiac disease and autoimmune disease but not significant in our sample population |
|

Reviewer 4 Report
Thank you for making the requested changes.
Please complete the introduction part with relevant data related to anxiety, depression and stress of chronically ill patients and depression, anxiety and stress related to COVID-19, each of them separately and see what happens when put together in your study.
Please delete the part in introduction about compassion fatigue at professionals, it is not relevant for your study.
Author Response
|
|
||
|
Comment |
Author Response |
Text Insertion (if applicable)/ section
|
|
Please complete the introduction part with relevant data related to anxiety, depression and stress of chronically ill patients and depression, anxiety and stress related to COVID-19, each of them separately and see what happens when put together in your study.
|
Line 48-60 Chronic illnesses have a high death rate and are quite costly on healthcare infrastructure [1]. The World Health Organization (WHO) estimates that in 2020, chronic diseases accounted for 60% of all disease burden worldwide and 73% of all fatalities. In addition, developing nations will account for 79% of these deaths[2]. Previous studies demonstrates the high rates of stress, anxiety, and depression in chronic disease patients. It is advised that health professionals focus more on preventing and controlling these illnesses [3]. Studies reported more signs of anxiety and more stress in people with chronic disease than in those without any chronic disease during Covid-19pandemic [4]. Protecting older people's mental health is crucial, especially for those who have chronic illnesses. In particular, in these difficult times that we are presently experiencing, it is necessary to give these vulnerable segments of the population with psychological interventions and instruments aimed at enhancing their emotional and social states [5].
|
|
|
Please delete the part in introduction about compassion fatigue at professionals, it is not relevant for your study |
Done |
|
